# HYBRID REPRESENTATION LEARNING VIA EPISTEMIC GRAPH

## ABSTRACT

In recent years, deep models have achieved remarkable success in many vision tasks. Unfortunately, their performance largely depends on intensive training samples. In contrast, human beings typically perform hybrid learning, e.g., spontaneously integrating structured knowledge for cross-domain recognition or on a much smaller amount of data samples for few-shot learning. Thus it is very attractive to extend hybrid learning for the computer vision tasks by seamlessly integrating structured knowledge with data samples to achieve more effective representation learning. However, such a hybrid learning approach remains a great challenge due to the huge gap between the structured knowledge and the deep features (learned from data samples) on both dimensions and knowledge granularity. In this paper, a novel Epistemic Graph Layer (EGLayer) is developed to enable hybrid learning, such that the information can be exchanged more effectively between the deep features and a structured knowledge graph. Our EGLayer is composed of three major parts: (a) a local graph module to establish a local prototypical graph through the learned deep features, i.e., aligning the deep features with the structured knowledge graph at the same granularity; (b) a query aggregation model to aggregate useful information from the local graphs, and using such representations to compute their similarity with global node embeddings for final prediction; and (c) a novel correlation alignment loss function to constrain the linear consistency between the local and global adjacency matrices from both cosine similarity and Euclidean space. EGLayer is a plug-and-play module that can replace the standard linear classifier, significantly improving the performance of deep models. Extensive experiments have demonstrated that EGLayer can greatly enhance representation learning for the tasks of cross-domain recognition and few-shot learning, and the visualization of knowledge graphs can aid in model interpretation.

## 1 INTRODUCTION

Over the past decade, deep models have attained significant achievements in many vision tasks. However, their success largely hinges on huge amounts of data samples and complicated model architectures Bian et al. (2014); Chen & Lin (2014); De Bézenac et al. (2019); Xie et al. (2021). In contrast, human beings typically perform hybrid learning, e.g., spontaneously integrating structured knowledge with a much smaller amount of data samples to achieve more effective representation learning, and can thus easily achieve cross-domain recognition. Therefore, it is very attractive to extend such hybrid learning for the computer vision tasks by seamlessly integrating structured knowledge with data samples to achieve more effective representation learning. However, such a hybrid learning approach remains a great challenge due to the huge gap between the structured knowledge and the deep features (learned from data samples) on both dimensions and knowledge granularity.

Graph has provided a direct and intuitive approach to represent structured knowledge. In a graph, each node represents one specific entity, while the relationship between the entities is represented by the edge adjacency matrix. Compared with conventional knowledge fusion methods Hu et al. (2016); Allamanis et al. (2017); Kodirov et al. (2017); Bansal et al. (2018); Gürel et al. (2021); Badreddine et al. (2022), graph-based methods have two distinct advantages: 1. Node embeddings could represent general concept of an entity with ample knowledge; 2. By concentrating on relational adjacency matrix, the graph representation is intuitively closer to humans' structured knowledge.

One critical challenge for hybrid learning (e,g., incorporating knowledge graph into data-driven deep learning) is the mismatch between deep features (learned from data samples) and graph representations of structured knowledge. Such mismatch can be divided into two parts: firstly, the deep features typically represent the visual distribution of a single image, while the structured graph contains the overall semantic knowledge which commonly share among many images, i.e., their information granularities are significantly different. Secondly, the deep features are usually in high dimensions, while the structured knowledge graph is a set of nodes and edges with much lower dimensions. Existing methods Lee et al. (2018); Liang et al. (2018); Chen et al. (2019; 2020); Naeem et al. (2021) mostly rely on a simple linear mapping or matrix multiplication to merge them, which could be ineffective and unstable.

To address the mismatch issue on information granularity, we propose a hybrid learning method with a local graph module, which establishes dynamic update of a local prototypical graph by historical deep features. This module acts as a memory bank and transfers deep features to the holistic visual graph. We then devise a query aggregation model that injects the current deep feature to the local graph and uses a Graph Neural Network (GNN) Kipf & Welling (2016); Hamilton et al. (2017); Veličković et al. (2017) to aggregate information for both the local graph node and feature node, aligning them to the same dimension as the global graph. The final prediction is performed based on the similarity between the knowledge-enhanced deep features and the global node embeddings. Moreover, a novel correlation alignment loss function is developed to maintain linear consistency between the local graph and the global one by constraining the adjacency matrix from both cosine similarity and Euclidean space. Together, these three parts constitute a well functional Epistemic Graph Layer (EGLayer).

The EGLayer is a versatile plug-and-play module that can be easily integrated into most existing deep models in the place of a standard linear classifier. Our experiments on the computer vision tasks of cross-domain recognition and few-shot learning have demonstrated the effectiveness of our proposed hybrid learning approach with such EGLayer, which achieves significant improvements on both performance and robustness. Moreover, such EGLayer has also shown promising results compared with conventional knowledge integration methods. Finally, the visualization of the local graphs and the global ones can provide valuable insights for model interpretation.

## 2 RELATED WORKS

Research on integrating human knowledge into deep models using graphs has drawn significant attention in recent years, which can be mainly categorized into two streams: visual guided graph representation learning and knowledge graph guided visual feature learning.

### 2.1 VISUAL GUIDED GRAPH REPRESENTATION LEARNING

In this direction, Wang et al. (2018); Chen et al. (2019); Gao et al. (2019); Kampffmeyer et al. (2019); Peng et al. (2019); Chen et al. (2020) often involve taking a fixed visual feature extractor and designing a function to transform graph embeddings to visual features, which are then fused together. Wang et al. (2018) construct a Graph Convolutional Network (GCN) using the WordNet structure and train it to predict visual classifiers pre-trained on ImageNet. By leveraging the relationships learned by GCN, it is able to transfer knowledge to new class nodes and perform zero-shot learning. Subsequently, Peng et al. (2019) improve their works by presenting knowledge transfer network, which replaces the inner product with cosine similarity of images. Moreover, Chen et al. (2020) propose a knowledge graph transfer network, which freezes the visual feature extractor and adopts three distance metrics to measure the similarity of visual features.

### 2.2 KNOWLEDGE GRAPH GUIDED VISUAL FEATURE LEARNING

Other works Socher et al. (2013); Norouzi et al. (2013); Zhang & Saligrama (2015); Liang et al. (2018); Monka et al. (2021); Radford et al. (2021) commonly focus on knowledge graph guided visual feature learning, as knowledge graph is considered more reliable than visual features. These works usually treat the knowledge graph as either a fixed external knowledge base or a high-level supervision to visual features. Socher et al. (2013) utilize a combination of dot-product similarity and hinge rank loss to learn a linear transformation function between the visual embedding space and the

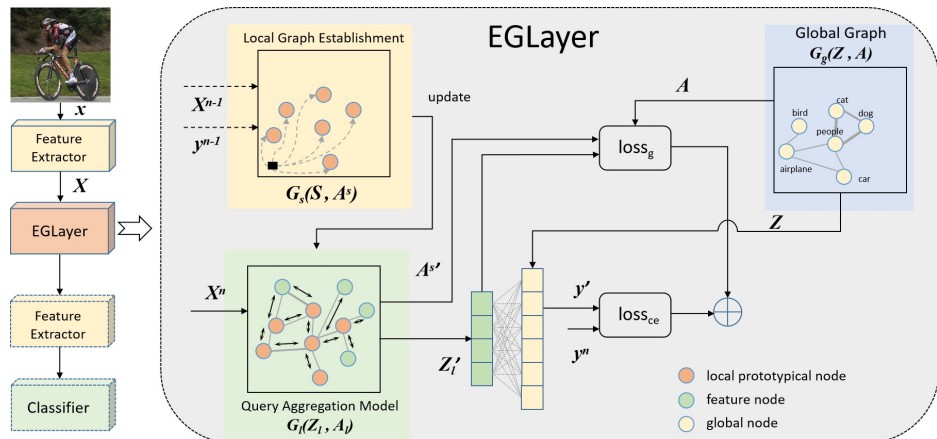

Figure 1: This figure illustrates the general framework of our proposed Epistemic Graph Layer. It can be inserted after any feature extractor layer to transfer the image feature dimension and granularity. In this paper, we primarily focus on replacing the standard linear classifier.

semantic embedding space to avoid the fails in high dimensional space. More important, Zhang & Saligrama (2015) propose a semantic similarity embedding method by representing target instances as a combination of a proportion of seen classes. They create a semantic space where each novel class is expressed as a probability mixture of the projected source attribute vectors of the known classes. Recently, Monka et al. (2021) adopt knowledge graph to train a visual feature extractor by a contrastive knowledge graph embedding loss, which outperforms conventional methods.

To our best knowledge, existing works hardly make efforts on aligning the knowledge granularity between local image features and the global graph. Consequently, they usually suffer from inefficient knowledge fusion issues and under-utilize the knowledge embedded in the graph. Such an observation drives us to explore a reliable and flexible knowledge graph projection method.

## 3 METHOD

For a typical classification task, we have a dataset $\mathcal{D} = (\boldsymbol{x}, \boldsymbol{y})$ to train a model, where $\boldsymbol{x}$ represents input images and $\boldsymbol{y}$ are their respective labels. Firstly, we exploit a feature extractor $f_\theta$ to extract image features $\boldsymbol{X} \in \mathbb{R}^D$ from $\boldsymbol{x}$, where $\theta$ represents the learnable parameters. Then, a classifier is utilized to compute the probability of each category given the feature. Finally, the loss function (commonly used cross-entropy loss) between $\boldsymbol{y}'$ and $\boldsymbol{y}$ are ultilized for optimization:

$$\boldsymbol{X} = f_\theta(\boldsymbol{x}), \quad \boldsymbol{y}' = \boldsymbol{W}\boldsymbol{X}, \quad \mathcal{L}_{sup} = loss_{ce}(\boldsymbol{y}, \boldsymbol{y}'). \tag{1}$$

Except for the labels of each instance, there are additional knowledge graph available during model training. Assuming we have a global knowledge graph $\boldsymbol{G}_g$, the critical problem is how to integrate it to facilitate model training. We define $\boldsymbol{G}_g = (\boldsymbol{Z}, \boldsymbol{A})$, where $\boldsymbol{Z} \in \mathbb{R}^{n \times d}$ represents the $n$ nodes with $d$-dimensional features, and $\boldsymbol{A} \in \mathbb{R}^{n \times n}$ denotes the edges among the $n$ nodes.

### 3.1 LINEAR PROJECTION LAYER

To integrate the knowledge graph $\boldsymbol{G}_g$ into model training, the first step is to project the visual features to the same dimension as the graph nodes. The most straightforward approach is using a linear layer, where $\boldsymbol{W}_p \in \mathbb{R}^{d \times D}$ denotes the learnable mapping matrix. Next, we can calculate the cosine similarity between $\boldsymbol{Z}'$ and the global graph node embedding $\boldsymbol{Z}_i$ to get final prediction $\boldsymbol{y}'$, where $\langle \cdot, \cdot \rangle$ represents the cosine similarity of two vectors. The overall formulations are:

$$\boldsymbol{X} = f_\theta(\boldsymbol{x}), \quad \boldsymbol{Z}' = \boldsymbol{W}_p\boldsymbol{X}, \quad \boldsymbol{y}' = \frac{\exp\left(\langle \boldsymbol{Z}', \boldsymbol{Z}_i \rangle\right)}{\sum_n \exp\left(\langle \boldsymbol{Z}', \boldsymbol{Z}_i \rangle\right)}, \quad \mathcal{L}_{sup} = loss_{ce}(\boldsymbol{y}, \boldsymbol{y}'). \tag{2}$$

## 3.2 EPISTEMIC GRAPH LAYER

To achieve an efficient interaction between local features and prior knowledge, we propose a novel EGLayer, which is composed of three major parts. Firstly, the local graph module establishes a dynamically updated prototypical graph by historical features. This module transfers the instance-level features to the graph-level representation, acting as a memory bank. Secondly, in the query aggregation model, the extracted features are injected into the obtained local graph to generate the query graph, which is further fed into a GNN to aggregate information for both feature and local graph nodes. In this manner, we fulfill a natural dimension alignment between the local and global graphs and obtain the prediction logits. Finally, we propose an auxiliary correlation alignment loss by constraining the local and global correlation adjacency matrices, further ensuring linear consistency and comparable knowledge granularity between the local and global graphs from cosine and Euclidean perspectives. The overall framework is shown in Figure 1.

### 3.2.1 LOCAL GRAPH ESTABLISHMENT

To align visual features with the global graph, we first establish a local graph $G_l = (Z_l, A_l)$ by the extracted features. We define $\mathcal{D}_k$ as the set of $k$-th category samples. The local prototype can be obtained by averaging the features of the categories:

$$S'_k = \frac{1}{|\mathcal{D}_k|} \sum_{(\boldsymbol{x}_i, \boldsymbol{y}_i) \in \mathcal{D}_k} f_\theta(\boldsymbol{x}_i) \tag{3}$$

To dynamically maintain the local prototype, we leverage exponential moving average scheme Cai et al. (2021); Huang et al. (2021); Xu et al. (2022) to update $S_k \in \mathbb{R}^D$ in each iteration:

$$S_k = \beta S_k + (1 - \beta) S'_k \tag{4}$$

The local prototype $S$ represents the node embeddings of the local graph, which serves as a memory bank that preserves historical visual features, aligning the granularity of the local graph with the semantic global graph. To facilitate the interaction between the extracted feature and local graph, we build the updated local graph embedding $Z_l$:

$$Z_l = [\underbrace{S_1 S_2 \cdots S_n}_{\text{local prototypes}} \underbrace{X_1 X_2 \cdots X_q}_{\text{query samples}}]^{\mathrm{T}}. \tag{5}$$

### 3.2.2 QUERY AGGREGATION MODEL

To align the local graph with global graph in the same dimensional space, we utilize GNNs via the aggregation operator. Before the aggregation process, we need to define the adjacency matrix $A_l$. For each local prototype $S$ in the $G_l$, it is expected to aggregate information from highly related local graph nodes. We compute the adjacency matrix $A^s$ using the Gaussian kernel $\mathcal{K}_G$ Liu et al. (2019); Wang et al. (2020a); Xu et al. (2022):

$$A^s = (a^s_{ij}) \in \mathbb{R}^{n \times n} = \mathcal{K}_G(S_i^{\mathrm{T}}, S_j^{\mathrm{T}}) = \exp\left(-\frac{\|S_i^{\mathrm{T}} - S_j^{\mathrm{T}}\|_2^2}{2\sigma^2}\right), \tag{6}$$

where $\sigma$ is a hyperparameter to control the sparsity of $A^s$ that is set as 0.05 by default. Moreover, $A^s$ is a symmetric matrix ($a^s_{ij} = a^s_{ji}$), so each node can both aggregate and transfer information.

The query node $X$ also needs to aggregate useful information from the prototypical nodes, and the unidirectional aggregation matrix $A^{xs}$ is defined as:

$$A^{xs} = (a^{xs}_{ij}) \in \mathbb{R}^{n \times q} = \mathcal{K}_G(S_i^{\mathrm{T}}, X_j^{\mathrm{T}}) = \exp\left(-\frac{\|S_i^{\mathrm{T}} - X_j^{\mathrm{T}}\|_2^2}{2\sigma^2}\right). \tag{7}$$

Subsequently, we calculate the adjacency matrix $\boldsymbol{A}_l$:

$$\boldsymbol{A}_l = \begin{bmatrix} \boldsymbol{A}^s & \boldsymbol{A}^{xs} \\ \boldsymbol{A}^{xs\mathrm{T}} & \boldsymbol{E} \end{bmatrix}, \tag{8}$$

where $\boldsymbol{E}$ is the identity matrix since query features are not allowed to interact with each other.

With the local graph embedding $\boldsymbol{Z}_l$ and adjacency matrix $\boldsymbol{A}_l$, we exploit GCN Estrach et al. (2014); Kipf & Welling (2016) to conduct the aggregation operation:

$$\boldsymbol{H}^{(m+1)} = \sigma \left( \tilde{\boldsymbol{D}}_l^{-\frac{1}{2}} \tilde{\boldsymbol{A}}_l \tilde{\boldsymbol{D}}_l^{-\frac{1}{2}} \boldsymbol{H}^{(m)} \boldsymbol{W}^{(m)} \right), \tag{9}$$

where $\tilde{\boldsymbol{A}}_l$ is the local correlation matrix $\boldsymbol{A}_l$ with self-connections, and $\tilde{\boldsymbol{D}}_l$ is the degree matrix of $\tilde{\boldsymbol{A}}_l$. $\boldsymbol{W}^{(m)}$ denotes the learnable matrix in $m$-th layer, while $\sigma$ is the activation function. Here, we take the local graph embedding $\boldsymbol{Z}_l$ as the first layer input of $\boldsymbol{H}^{(m)}$, and the final aggregated node representation $\boldsymbol{H}^{(m+1)}$ are defined as $\boldsymbol{Z}_l'$.

Finally, we exploit Eq. 2 to calculate the final predictions by $\boldsymbol{Z}_l'$ and global node embedding $\boldsymbol{Z}$.

### 3.2.3 CORRELATION ALIGNMENT LOSS

To obtain adequate and consistent guidance by global graph, we here intentionally constrain the local adjacency matrix. However, the local adjacency matrix is fixed in each training iteration since $\boldsymbol{A}^s$ is only related to the local graph embedding $\boldsymbol{S}$, which is updated in advance of each iteration. Therefore, we introduce an extra trainable matrix $\boldsymbol{W}_a$ for $\boldsymbol{A}^s$ to get the amended adjacency matrix:

$$\boldsymbol{A}^{s\prime} = \left( a_{ij}^{s\prime} \right) \in \mathbb{R}^{n \times n} = \boldsymbol{W}_a \boldsymbol{A}_{i,j}^s = \boldsymbol{W}_a \mathcal{K}_G \left( \boldsymbol{S}_i^{\mathrm{T}}, \boldsymbol{S}_j^{\mathrm{T}} \right). \tag{10}$$

Then, the adjacency matrix $\boldsymbol{A}^s$ in Eq. 8 can be replaced by $\boldsymbol{A}^{s\prime}$. Accordingly, we can build an auxiliary loss function by optimize $\boldsymbol{A}^{s\prime}$ to the global adjacency matrix $\boldsymbol{A}$:

$$\mathcal{L}_a(\boldsymbol{A}, \boldsymbol{A}^{s\prime}) = -\frac{1}{n^2} \sum_{i=1}^{n} \sum_{j=1}^{n} a_{ij} \log \left( \sigma \left( a_{ij}^{s\prime} \right) \right) + (1 - a_{ij}) \log \left( 1 - \sigma \left( a_{ij}^{s\prime} \right) \right), \tag{11}$$

where $\sigma(\cdot)$ is sigmoid function and $\mathcal{L}_a$ could be viewed as a binary cross-entropy loss for each correlation value with soft labels.

Moreover, since $\boldsymbol{A}$ and $\boldsymbol{A}^{s\prime}$ both come from Euclidean space, we design a new regularization term based on cosine similarity to make learned embedding $\boldsymbol{S}$ more distinctive. The regularization and the final loss are calculated as:

$$\mathcal{L}_{reg}(\boldsymbol{S}) = \| \langle \boldsymbol{S}, \boldsymbol{S}^{\mathrm{T}} \rangle \|_2 = \|\boldsymbol{C}\|_2 = \| (c_{ij}) \in \mathbb{R}^{n \times n} \|_2 = \sqrt{\sum_{i=1}^{n} \sum_{j=1}^{n} c_{ij}^2} \tag{12}$$

$$\mathcal{L} = \mathcal{L}_{sup} + \alpha \mathcal{L}_g = \mathcal{L}_{sup} + \alpha_1 \mathcal{L}_a + \alpha_2 \mathcal{L}_{reg} \tag{13}$$

## 4 EXPERIMENTS

As discussed before, our proposed EGLayer is a plug-and-play module that can benefit most kinds of deep models by conveniently replacing their standard classifiers. In order to examine the effectiveness of our knowledge guidance and extrapolation, we mainly focused on several challenging tasks, including cross-domain classification, universal domain adaptation, and few-shot learning.

The establishment of the global knowledge graph has various available scheme. The co-occurrence graph Duque et al. (2018); Chen et al. (2019); Wang et al. (2020b) represents the frequency of two classes occurring together, but is not suitable for single-label tasks and heavily depends on the size

Table 1: Comparison experiments on Office-31 dataset

| Methods | A→W | D→W | W→D | A→D | D→A | W→A | Average |
|---|---|---|---|---|---|---|---|
| ResNet50 | 65.41 | 79.25 | 91.00 | 70.00 | 44.68 | 50.38 | 66.79 |
| ResNet50 + LPLayer | 67.92 | 85.53 | 94.00 | 71.00 | 53.62 | 56.22 | 71.38 |
| ResNet50 + EGLayer | **70.44** | **90.57** | **96.00** | **77.00** | **56.96** | **57.87** | **74.81** |

Table 2: Comparison experiments on Office-Home dataset

| Methods | A→C | A→P | A→R | C→A | C→P | C→R | P→A | P→C | P→R | R→A | R→C | R→P | Average |
|---|---|---|---|---|---|---|---|---|---|---|---|---|---|
| ResNet50 | 40.42 | **59.48** | **69.10** | 45.07 | **56.55** | **60.13** | 39.71 | 39.86 | 68.09 | 58.64 | 43.60 | 73.64 | 54.52 |
| ResNet50 + LPLayer | 40.78 | 28.69 | 66.43 | 40.48 | 32.88 | 43.04 | 56.68 | **44.81** | **69.03** | 65.08 | **49.79** | 52.58 | 49.19 |
| ResNet50 + EGLayer | **41.81** | 57.95 | 65.74 | **53.36** | 53.35 | 56.34 | **62.52** | 41.67 | 68.28 | **70.33** | 45.54 | **73.67** | **57.55** |

of the dataset. Another option is the pre-defined knowledge graph Lin et al. (2015); Toutanova et al. (2016); Krishna et al. (2017), which is constructed using manually labeled relational datasets or knowledge base. In our approach, we adopt a simpler solution by utilizing the word embeddings from GloVe Pennington et al. (2014) and Eq. 6 to obtain node embeddings and adjacency matrices, which is an adaptive approach that does not require additional sources of knowledge.

Importantly, in our experiments, we solely utilize class information from the training set without incorporating any novel classes into the global knowledge graph. In the context of open-set domain adaptation, our approach initiates with training the model on the source domain, focusing on source classes. We subsequently apply a threshold to filter out images that do not belong to known classes within the source domain, classifying them as outlier classes. In the realm of few-shot learning, our method trains the feature extractor and constructs both global and local graphs based on the base classes. In validation and testing phases, we employ the trained feature extractor to extract image features for the few-shot images belonging to the novel class. Afterwards, unlabeled test images are compared to these few-shot features using cosine similarity to determine their respective classes.

## 4.1 CROSS-DOMAIN CLASSIFICATION

### 4.1.1 DATASETS

In this experiment, we train the model on the source domain and then perform classification directly on the target domain without using any target domain data. We conduct experiments on two datasets, namely Office-31 Saenko et al. (2010) and Office-Home Venkateswara et al. (2017). The Office-31 dataset consists of 4,652 images from 31 categories and is divided into three domains: *Amazon* (A), *Dslr* (D), and *Webcam* (W). The Office-Home dataset has 15,500 images with 65 categories and is divided into four domains: *Art* (A), *Clipart* (C), *Product* (P), and *Real World* (R).

### 4.1.2 COMPARISON RESULTS

Table 1 and Table 2 present the results of our experiments with different model settings. ResNet50 He et al. (2016) refers to ResNet50 backbone with a standard linear classifier. ResNet50 + LPLayer denotes the ResNet50 backbone with the linear projection layer described in Section 3.1. ResNet50 + EGLayer is the ResNet50 backbone equipped with our proposed epistemic graph layer. The only difference among the three models is the classifier, which allows us to make a fair comparison.

On average, ResNet50 + LPLayer outperforms ResNet50 by 4.59% on Office-31, and ResNet50 + EGLayer further yields a 3.43% performance gain and obtains the best results in all cases. Unexpectedly, ResNet50 + LPLayer shows an obvious performance drop on Office-Home by 5.33%, which could be attributed to its insufficient knowledge integration. In contrast, ResNet50 + EGLayer achieves a remarkable improvement by 3.03%. Specifically, the largest margin is reported in the D→W task on Office-31, where ResNet50 + EGLayer improves the results from 79.25% to 90.57%, an impressive increase of 11.32%. These results suggest the EGLayer learns a better representation.

### 4.1.3 VISUALIZATION OF GRAPHS

We visualize two graphs including enhanced local graph and global graph. To show the results clearly, we only show the top-150 edges of strong relationship, and the thicker edge represents the higher relational edge value (See Appendix for more details).

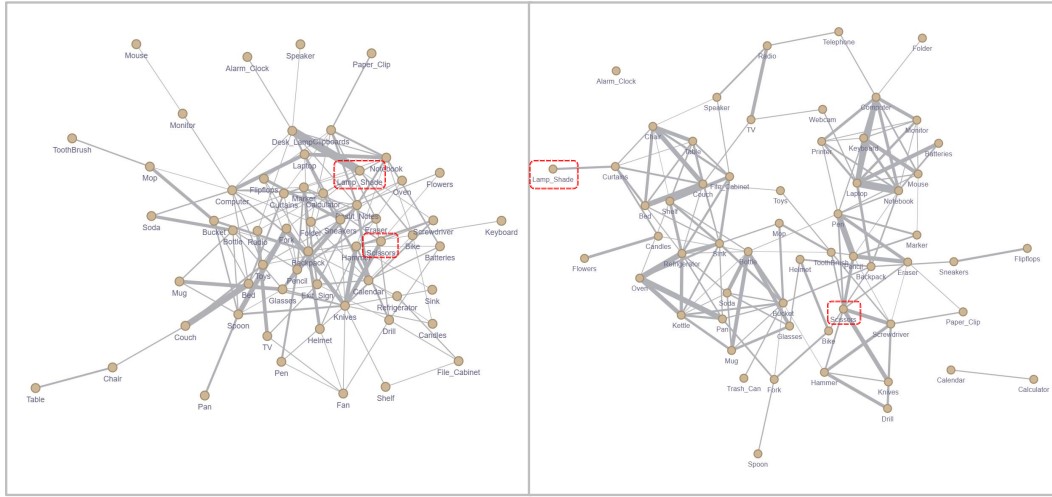

Figure 2: The left visualized graph is enhanced local graph, and the right is global graph. These experiments are conducted in Office-Home datasets of 65 classes in *Clipart* domain. We have highlighted two typical nodes: the *Lamp Shade* is visually similar to *Desk Lamp* while the *Scissors* is semantically closer to stationery objects.

The enhanced local graph mainly contains knowledge from visual sources, while the global graph consists of more semantic knowledge. As shown in Figure 2, we highlight two typical nodes. The *Scissors* node in the global graph is close to two types of concepts, tools and stationeries. The typical tools include *Knives*, *Hammer*, and *Screwdriver*, while the stationeries contain *Eraser*, *Pencil*, and *Pen*. In the enhanced local graph, *Scissors* is only related to the typical tools category due to their similar metallic appearance. Another interesting node is *Lamp Shade*, which is highly related to the *Desk Lamp* since the *Lamp Shade* image is always combined with the lamp. On the contrary, these two nodes do not have an edge in the global graph, which may be attributed to the semantic emphasis of *Lamp Shade* as a shade rather than a lamp.

## 4.2 OPEN-SET DOMAIN ADAPTATION

### 4.2.1 IMPLEMENTATION DETAILS

Table 3: Universal domain adaptation experiments on Office-31 dataset

| Methods | A→W | D→W | W→D | A→D | D→A | W→A | Average |
|---|---|---|---|---|---|---|---|
| DANN Ganin et al. (2016) | 80.65 | 80.94 | 88.07 | 82.67 | 74.82 | 83.54 | 81.78 |
| RTN Long et al. (2016) | **85.70** | 87.80 | 88.91 | 82.69 | 74.64 | 83.26 | 84.18 |
| IWAN Zhang et al. (2018) | 85.25 | 90.09 | 90.00 | 84.27 | 84.22 | 86.25 | 86.68 |
| PADA Zhang et al. (2018) | 85.37 | 79.26 | 90.91 | 81.68 | 55.32 | 82.61 | 79.19 |
| ATI Panareda Busto & Gall (2017) | 79.38 | 92.60 | 90.08 | 84.40 | 78.85 | 81.57 | 84.48 |
| OSBP Saito et al. (2018) | 66.13 | 73.57 | 85.62 | 72.92 | 47.35 | 60.48 | 67.68 |
| UAN You et al. (2019) | 77.16 | **94.54** | **95.48** | 78.71 | 84.47 | 82.14 | 85.42 |
| UAN + LPLayer | 83.69 | 91.20 | 95.17 | 84.90 | 84.93 | 84.24 | 87.36 |
| UAN + EGLayer | 83.51 | 94.23 | 94.34 | **86.11** | **87.88** | **88.26** | **89.06** |

In this subsection, we conduct experiments on open-set domain adaptation tasks, where the source and target domains have some shared and some private categories. We adopt the task definition proposed in You et al. (2019). Specifically, we denote the label sets of the source and target domains as $\mathcal{C}_s$ and $\mathcal{C}_t$, respectively, and $\mathcal{C} = \mathcal{C}_s \cap \mathcal{C}_t$ represents the set of shared categories. Furthermore, $\overline{\mathcal{C}}_s = \mathcal{C}_s \backslash \mathcal{C}$ and $\overline{\mathcal{C}}_t = \mathcal{C}_t \backslash \mathcal{C}$ represent the private categories in the source and target domains, respectively. We can then quantify the commonality between the two domains as $\xi = \frac{|\mathcal{C}_s \cap \mathcal{C}_t|}{|\mathcal{C}_s \cup \mathcal{C}_t|}$.

For the Office-31, we choose 10 categories as shared categories $\mathcal{C}$, the following 10 categories as source private categories $\mathcal{C}_s$, and the remaining categories as target private categories $\mathcal{C}_t$. For the

Table 4: Universal domain adaptation experiments on Office-Home dataset

| Methods | A→C | A→P | A→R | C→A | C→P | C→R | P→A | P→C | P→R | R→A | R→C | R→P | Average |
|---------|-----|-----|-----|-----|-----|-----|-----|-----|-----|-----|-----|-----|---------|
| DANN Ganin et al. (2016) | 56.17 | 81.72 | 86.87 | 68.67 | 73.38 | 83.76 | 69.92 | 56.84 | 85.80 | 79.41 | 57.26 | 78.26 | 73.17 |
| RTN Long et al. (2016) | 50.46 | 77.80 | 86.90 | 65.12 | 73.40 | 85.07 | 67.86 | 45.23 | 85.50 | 79.20 | 55.55 | 78.79 | 70.91 |
| IWAN Zhang et al. (2018) | 52.55 | 81.40 | 86.51 | 70.58 | 70.99 | 85.29 | 74.88 | 57.33 | 85.07 | 77.48 | 59.65 | 78.91 | 73.39 |
| PADA Zhang et al. (2018) | 39.58 | 69.37 | 76.26 | 62.57 | 67.39 | 77.47 | 48.39 | 35.79 | 79.60 | 75.94 | 44.50 | 78.10 | 62.91 |
| ATI Panareda Busto & Gall (2017) | 52.90 | 80.37 | 85.91 | 71.08 | 72.41 | 84.39 | 74.28 | 57.84 | 85.61 | 76.06 | 60.17 | 78.42 | 73.29 |
| OSBP Saito et al. (2018) | 47.75 | 60.90 | 76.78 | 59.23 | 61.58 | 74.33 | 61.67 | 44.50 | 79.31 | 70.59 | 54.95 | 75.18 | 63.90 |
| UAN You et al. (2019) | 65.92 | 79.82 | 88.09 | 71.99 | 75.11 | 84.54 | 77.56 | **64.16** | 89.06 | **81.92** | 65.87 | 83.80 | 77.32 |
| UAN + LPLayer | **67.43** | 81.64 | 88.97 | 76.19 | 81.58 | 87.29 | 79.86 | 63.11 | 88.73 | 79.70 | **68.62** | 84.07 | 78.93 |
| UAN + EGLayer | 66.47 | **84.53** | **92.36** | **80.97** | **82.79** | **89.40** | **80.12** | 63.35 | **91.98** | 79.48 | 64.54 | **85.43** | **80.12** |

Office-Home, we take the first 10 categories as $\mathcal{C}$, the next 5 categories $\mathcal{C}_s$, and the rest as $\mathcal{C}_t$. As a result, we obtain $\xi$ values of 0.32 and 0.15 for the Office-31 and Office-Home, respectively. (See Appendix for more experiments)

Table 5: Comparison with state-of-the-art methods on miniImageNet dataset.

| Methods | Backbone | 1-shot | 5-shot |
|---------|----------|--------|--------|
| SNAIL Mishra et al. (2017) | ResNet-12 | 55.71 ± 0.99 | 68.88 ± 0.92 |
| AdaResNet Munkhdalai et al. (2018) | ResNet-12 | 56.88 ± 0.62 | 71.94 ± 0.57 |
| TADAM Oreshkin et al. (2018) | ResNet-12 | 58.50 ± 0.30 | 76.70 ± 0.30 |
| MTL Sun et al. (2019) | ResNet-12 | 61.20 ± 1.80 | 75.50 ± 0.80 |
| MetaOptNet Lee et al. (2019) | ResNet-12 | 62.64 ± 0.61 | 78.63 ± 0.46 |
| ProtoNets + TRAML Li et al. (2020) | ResNet-12 | 60.31 ± 0.48 | 77.94 ± 0.57 |
| BOIL Oh et al. (2021) | ResNet-12 | - | 71.30 ± 0.28 |
| DAM Zhou et al. (2022) | ResNet-12 | 60.39 ± 0.21 | 73.84 ± 0.16 |
| Matching Networks Vinyals et al. (2016) | ConvNet-4 | 45.73 ± 0.19 | 57.80 ± 0.18 |
| Matching Networks + LPLayer | ConvNet-4 | 47.87 ± 0.19 | 57.84 ± 0.18 |
| Matching Networks + EGLayer | ConvNet-4 | **50.48 ± 0.20** | **61.29 ± 0.17** |
| Prototypical Networks Snell et al. (2017) | ConvNet-4 | 49.45 ± 0.20 | 66.38 ± 0.17 |
| Prototypical Networks + LPLayer | ConvNet-4 | 49.67 ± 0.20 | 66.66 ± 0.17 |
| Prototypical Networks + EGLayer | ConvNet-4 | **50.30 ± 0.20** | **67.88 ± 0.16** |
| Classifier-Baseline Chen et al. (2021) | ResNet-12 | 58.91 ± 0.23 | 77.76 ± 0.17 |
| Classifier-Baseline + LPLayer | ResNet-12 | 60.96 ± 0.23 | 78.07 ± 0.17 |
| Classifier-Baseline + EGLayer | ResNet-12 | **61.53 ± 0.27** | **78.84 ± 0.21** |
| Meta-Baseline Chen et al. (2021) | ResNet-12 | 63.17 ± 0.23 | 79.26 ± 0.17 |
| Meta-Baseline + LPLayer | ResNet-12 | 62.27 ± 0.23 | 77.63 ± 0.17 |
| Meta-Baseline + EGLayer | ResNet-12 | **63.55 ± 0.26** | **79.78 ± 0.54** |

### 4.2.2 COMPARISON RESULTS

We summarize the results in Table 3 and Table 4. To thoroughly understanding the effect of knowledge integration, we replace the linear classifier in UAN You et al. (2019) with LPLayer and EGLayer, namely UAN + LPLayer and UAN + EGLayer respectively.

In the open-world setting, integrating knowledge is a crucial factor of performance promotion. On average, UAN + LPLayer brings stable 1.94% and 1.61% improvements over vanilla UAN on Office-31 and Office-Home datasets. The proposed UAN + EGLayer further enhances the results by 1.70% and 1.19% in comparison to UAN + LPLayer, which demonstrates that EGLayer has a better generalization capability than conventional linear knowledge fusion. Interestingly, we observe that both knowledge-based methods obtain more significant improvements in challenging tasks (i.e. tasks with low accuracy), such as D→A and A→D. In general, UAN + EGLayer beats all competitors and reaches state-of-the-art performance in open-world setting.

### 4.3 FEW-SHOT LEARNING

### 4.3.1 DATASETS

We evaluate the few-shot learning task on two datasets. The miniImageNet Vinyals et al. (2016) is sampled from ImageNet Russakovsky et al. (2015) of 100 classes. 64 classes are used for training, the rest 16 and 20 classes are used for validation and testing, respectively. Each class contains 600

Table 6: Comparison with state-of-the-art methods on tieredImageNet dataset

| Methods | Backbone | 1-shot | 5-shot |
|---------|----------|--------|--------|
| MAML Finn et al. (2017) | ConvNet-4 | 51.67 ± 1.81 | 70.30 ± 1.75 |
| Relation Networks Sung et al. (2018) | ConvNet-4 | 54.48 ± 0.93 | 71.32 ± 0.78 |
| MetaOptNet Lee et al. (2019) | ResNet-12 | 65.99 ± 0.72 | 81.56 ± 0.53 |
| BOIL Oh et al. (2021) | ResNet-12 | 48.58 ± 0.27 | 69.37 ± 0.12 |
| DAM Zhou et al. (2022) | ResNet-12 | 64.09 ± 0.23 | 78.39 ± 0.18 |
| A-MET Zheng et al. (2023) | ResNet-12 | 69.39 ± 0.57 | 81.11 ± 0.39 |
| Matching Networks Vinyals et al. (2016) | ConvNet-4 | 41.99 ± 0.19 | 52.70 ± 0.19 |
| Matching Networks + LPLayer | ConvNet-4 | 42.61 ± 0.20 | 52.91 ± 0.19 |
| Matching Networks + EGLayer | ConvNet-4 | **45.87 ± 0.22** | **59.90 ± 0.19** |
| Prototypical Networks Snell et al. (2017) | ConvNet-4 | 48.65 ± 0.21 | 65.55 ± 0.19 |
| Prototypical Networks + LPLayer | ConvNet-4 | 48.97 ± 0.21 | 65.52 ± 0.19 |
| Prototypical Networks + EGLayer | ConvNet-4 | **50.17 ± 0.22** | **68.42 ± 0.18** |
| Classifier-Baseline Chen et al. (2021) | ResNet-12 | 68.07 ± 0.26 | 83.74 ± 0.18 |
| Classifier-Baseline + LPLayer | ResNet-12 | 68.28 ± 0.26 | 83.04 ± 0.18 |
| Classifier-Baseline + EGLayer | ResNet-12 | **69.38 ± 0.53** | **84.38 ± 0.59** |
| Meta-Baseline Chen et al. (2021) | ResNet-12 | 68.62 ± 0.27 | 83.74 ± 0.18 |
| Meta-Baseline + LPLayer | ResNet-12 | 69.16 ± 0.56 | 82.64 ± 0.41 |
| Meta-Baseline + EGLayer | ResNet-12 | **69.74 ± 0.56** | **83.94 ± 0.58** |

images resized to $84 \times 84$ resolution. The tieredImageNet Ren et al. (2018) is a larger datasets consisting of 608 classes sampled from ImageNet Russakovsky et al. (2015) too. All classes are divided 351, 97, 160 classes for training, validation and testing. Different from miniImageNet, tieredImageNet is more challenging owing to the long semantic distance between base and novel classes (See Appendix for more implementation details and experiments).

### 4.3.2 COMPARISON RESULTS

We compare our proposed method with mainstream methods in Table 5 and Table 6. All results are average of 5-way accuracy with 95% confidence interval. To verify our method's ability of lightweight plug-and-play modules, we implement our methods with four prevailing baselines Matching Networks Vinyals et al. (2016), Prototypical Networks Snell et al. (2017), Classifier-Baseline Chen et al. (2021), and Meta-Baseline Chen et al. (2021).

For miniImageNet, four LPLayer version holds a marginal improvement over the baseline, and inserting a LPLayer even causes a slight performance decline in Meta-Baseline. Conversely, EGLayer obtains stable improvements in all results. Especially for Matching Networks and Classifier-Baseline, EGLayer gains 4.75%/3.49% and 2.62%/1.08% promotion.

For tieredImageNet, compared with LPLayer, EGLayer enables a more adequate and reliable knowledge injection and achieves significant advantages in both settings. In detail, the EGLayer holds 3.88%/7.20%, 1.52%/2.87% improvements with Matching Networks and Prototypical Networks. For Classifier-Baseline and Meta-Baseline, EGLayer also have a remarkable advantages in 1-shot setting with 1.31% and 1.12% performance promotion.

## 5 CONCLUSIONS

In this paper, a novel EGLayer is introduced to enable hybrid learning, which can achieve more effective information exchange between the local deep features and a structured global knowledge graph. EGLayer is a plug-and-play module to replace the standard linear classifier, and it can significantly improve the performance of deep models by seamlessly integrating structured knowledge with data samples for deep learning. Our extensive experiments have demonstrated that our proposed hybrid learning approach with such EGLayer can greatly enhance representation learning for the tasks of cross-domain recognition and few-shot learning, and the visualization of knowledge graphs can aid in model interpretation effectively.

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
