# OpenReview forum: "Hybrid Representation Learning Via Epistemic Graph"
_ICLR.cc/2024/Conference — ICLR 2024 Conference Withdrawn Submission_

### Official Review · Reviewer_kGHn · 2023-10-24

**Soundness:** 3 good
**Presentation:** 1 poor
**Contribution:** 2 fair
**Rating:** 5
**Confidence:** 4

**Summary:**

This paper proposes a plug-and-play module called Epistemic Graph Layer(EGLayer) that can replace classifiers and consists of three main components: a local graph module, a query aggregation model, and a  correlation alignment loss function. The aim is to implement hybrid learning between visual image features and a priori knowledge graphs.

**Strengths:**

1. The proposed idea of hybrid learning of prototype and node local features is relatively novel.
2. The authors conducted a large number of experiments to prove the effectiveness of their method.

**Weaknesses:**

1. The description of the specific problem and the illustrated motivation are both not clear enough. Why does this work use the hybrid learning to solve the problem？
2. Details are poorly represented, e.g., the source of a priori knowledge graphs and the source of query samples. Due to the lack of some specific explanations, the formulation of the method makes it difficult to understand. Thus, the readers cannot basically reproduce this method.
3. The presentation of this manuscript need to be revised, for example:
- the 3rd sentence of the 5th paragraph of INTRODUCTION, the dimension alignment should put before the aggregation of hybrid local graphs;
- The first paragraph of section 3.2.2, "Moreover..." , whether a node can perform message passing is independent of whether it is an undirected graph or not.
4. The writing in this manuscript should be polished thoroughly, for example
- There is a lot of reuse of sentences (Abstract) in both INTRODUCTION and METHOD;
- Some descriptions about prototype in section 3.2.1 need be modified, e.g., the explanation for Eqn.(4).

**Questions:**

1. How to obtain a priori knowledge graph for an arbitrary collection of images?

2. Figure 2 shows inherent differences between a priori knowledge graph based on textual semantics and a dynamic prototype based on visual semantics. But Eq.2 in Section 3.1 indicates that the learned visual features are allowed to approximate the textual features of the a priori knowledge, does this compromise the learned visual semantic knowledge?

3. What does q in Eq.5 stand for? Is it the batch size? This will make the prototype nodes and the single-batch inner nodes placed in a single graph. Does only one mapping matrix preserve the consistency of their semantic space? This should be the core part of the work, could you provide some theoretical proof or more explanation?

**Details Of Ethics Concerns:**

NULL

---

### Official Review · Reviewer_EhZr · 2023-11-04

**Soundness:** 2 fair
**Presentation:** 2 fair
**Contribution:** 1 poor
**Rating:** 3
**Confidence:** 5

**Summary:**

This work aims to utilize the structured knowledge of the data samples for cross-domain and few-shot vision tasks. They develop a plug-and-play module for modelling the structured knowledge. The main idea consists of a local graph, the aggregation mechanism, and an alignment loss function. The authors conduct experiments on cross-domain recognition and few-shot classification to verify the effectiveness of the method.

**Strengths:**

1. The motivation is intuitive, introducing the structured knowledge is expected to be helpful for vision recognition tasks.
2. The overview of the framework (illustrated in Figure 1) is clear and easy-to-follow. The proposed method is a plug-and-play module and is easy to be incorporated into other frameworks.

**Weaknesses:**

1. The main concern is the novelty and rationality of the idea. The similar idea has been thoroughly explored in many previous works, like [A1][A2][A3][A4]. Please clarify the main difference and contribution of this work.
2. The experiments are significantly limited, and many recent works are missing. The latest baseline is Meta-Baseline, which was completed in 2021. Please remember that this submission is for ICLR 2024. Also, please verify your method with more powerful backbones (e.g., ResNet-50/101, ViT, etc.).
3. Why this idea is effective should be further explained and discussed.

[A1] Few-Shot Learning with Global Class Representations
[A2] Infinite Mixture Prototypes for Few-Shot Learning
[A3] Part-aware Prototype Network for Few-shot Semantic Segmentation
[A4] Local descriptor-based multi-prototype network for few-shot Learning

**Questions:**

Please check the weaknesses section.

---

### Official Review · Reviewer_rxaN · 2023-11-05

**Soundness:** 2 fair
**Presentation:** 1 poor
**Contribution:** 1 poor
**Rating:** 3
**Confidence:** 3

**Summary:**

This paper is inspired by the human's ability in leverage structured knowledge to efficiently learn new concepts with few samples or domain gaps. To endow the neural network with the same ability, the authors propose to use a new epistemic graph layer to align local (sample-wise) features with the global knowledge graph. Empirical evaluations are conducted on both domain adaptation and few-shot learning experiments.

**Strengths:**

1. The evaluations on multiple tasks and datasets are appreciated.

2. I like the idea of integrating explicit global knowledge into the neural network inference. However, it remains very unclear to me how the proposed method in this paper actually achieves this (see below).

**Weaknesses:**

My primary concern for this paper is the poor writing quality, which makes it very hard to evaluate the contribution of this paper.

Examples:

1. In the abstract, the authors discuss the very interesting motivation of mimicking human learning with structured knowledge. However, the rest of the abstract goes directly to how each module is constructed and 'communicates' with each other. What's the goal of each module, and how does each module fulfills the goal remain highly unclear. Concepts like 'global node' even appear without any proper explanation of the functionalities and motivations.

2. The last paragraph discusses the shortcomings of the existing methods without providing any support or evidence. What's the result of 'suffer from inefficient knowledge fusion issues and under-utilize the knowledge embedded in the graph' that the authors observed? This can hardly be an observation without presenting what the authors actually observed.

My low review confidence is mainly due to the fact that I believe a vast amount of necessary discussions is missing in this paper, making it hard to evaluate the contributions comprehensively.

**Questions:**

1. Neural network learn parameters across all layers which associatively convey knowledge about the learned task, and the feature propagation across layers can be considered as a form of information querying in an associative memory. What's the limitation of such global information that makes the proposed layer indispensable?

2. Please briefly explain how exactly the information granularities are aligned in the proposed layer. Are there any quantitative evaluations of the term 'granularity' that can support the claim?

3. Section 3.2, what does the 'prior knowledge' represent here? Is it equal to the global graph?

4. How many parameters does the proposed layer have compared to the standard linear layer? I believe many of the performance improvements are relatively marginal, especially for the ones with strong backbones, could the parameter size be a major reason of the improvement?

5. Please briefly explain how the proposed method aids model interpretation.